

# Discovering the fish fauna of a lagoon from the southeast of the Yucatan Peninsula, Mexico, using DNA barcodes

Adrian Emmanuel Uh-Navarrete[1], Martha Valdez-Moreno[1], Mariana E. Callejas-Jiménez[2] and Lourdes Vásquez-Yeomans[1]

[1] Department of Aquatic Ecology and Systematics, El Colegio de la Frontera Sur, Chetumal, Quintana Roo, Mexico
[2] Department of Observation and Study of the Land, the Atmosphere and the Ocean, El Colegio de la Frontera Sur, Chetumal, Quintana Roo, Mexico

## ABSTRACT

**Background**. Aquatic ecosystems in the tropics are typically environments with a high species richness of fishes. These systems are also among the most vulnerable in the world, threatening the overall biodiversity of tropical regions. As a first step, it is important to enumerate the species in any ecosystem to promote its conservation. This study aims to inventory the ichthyofauna in the Chile Verde Lagoon, Quintana Roo, on the Yucatan Peninsula, a system fortunately well protected in Mexico, based on faunal surveys backed up with mtDNA barcodes.

**Methods**. We collected larvae, juveniles, and adults of fishes in the lagoon with a variety of sampling gear targeting various life stages. Species were identified using both morphology and DNA barcodes. The abundance of species and ichthyoplankton biomass (wet weight, suction technique) were calculated from 43 samples.

**Results**. We collected 197 adult and juvenile fishes and 3,722 larvae, of which 306 specimens were DNA-sequenced with a success rate of 96.7%. We identified 13 families, 24 genera, and 27 species in our inventory. The species number was estimated to comprise 75% of the potential total richness using the Chao 1 richness estimator. Clupeids and gobiids accounted for 87.9% of the total abundance of fishes, and, together with cyprinodontids, also accounted for the highest ichthyoplankton biomass.

**Conclusion**. Adult and juvenile fishes were identified by morphology and meristic values, however larvae required DNA barcoding to identify species. The high biomass and abundance of larvae of clupeids, gobiids and cyprinodontids suggests that the Chile Verde Lagoon may be important for reproduction of these species in the region. *Microgobius microlepis*, a marine goby species, is reported for the first time in an inland oligohaline system. This study provides a basis for future environmental assessment and biomonitoring of the Chile Verde Lagoon in the Yucatan Peninsula of Mexico.

Corresponding author
Martha Valdez-Moreno,
mvaldez@ecosur.mx

## INTRODUCTION

The greatest diversity of fishes occurs in tropical regions (*Eschmeyer et al., 2010*; *Nelson, Grande & Wilson, 2016*). The species richness of inland fishes in a particular region is

augmented by the variety of available aquatic environments, from estuaries and lagoons to temporary ponds and underground rivers (*De Pinna, 2005*). Inland water ecosystems are among the most threatened environments in the world, with the great diversity found in the tropics especially vulnerable (*Dudgeon et al., 2006*; *Vörösmarty et al., 2010*). The fishes of inland waters are not exempt from this threat, especially due to habitat degradation (*Eby et al., 2005*), overexploitation (*Mulimbwa, Sarvala & Micha, 2019*), contamination with microplastics (*Silva-Cavalcanti et al., 2017*), and the invasion of exotic species (*Mazzoni et al., 2015*). Indeed, species are being lost worldwide, many disappearing before being discovered or described (*Costello, May & Stork, 2013*). An important first step to protect fish communities and prevent the loss of species is to inventory the fauna (*Montes-Ortiz & Elías-Gutiérrez, 2018*) and identify vulnerable species.

Most inventories of ichthyofauna focus on adult fishes but including egg and/or larval stages can significantly augment the species count (*e.g.*, *Schmitter-Soto et al., 2009*) by picking up species whose later stages are rare or difficult to collect. Early stages are often not assessed in inventories because these stages can be difficult to identify to species and require specialized taxonomists (*Ko et al., 2013*). This is especially true for many tropical inland water regions, where taxonomic keys are scarce and, in most cases, these stages have not been described (*Uh-Navarrete et al., 2021*).

DNA sequencing can overcome this hindrance, as a valuable method of identification linking early developmental stages to known species. Among these molecular tools, DNA barcoding, based on an extensive database of mtDNA COI sequences (*Hebert et al., 2003*), provides a fast and reliable way to identify species, independent of the stage of development. The effectiveness of identifications using this methodology has been widely demonstrated for adult fishes (*e.g.*, *Hubert et al., 2008*; *Valdez-Moreno et al., 2009*) as well as for early stages (*e.g.*, *Uh-Navarrete et al., 2021*). Nevertheless, there have been relatively few studies barcoding larvae of inland fishes (*Lira et al., 2022*), and only four in Mexico (*Elías-Gutiérrez et al., 2018*; *Montes-Ortiz & Elías-Gutiérrez, 2018*; *Uh-Navarrete et al., 2021*; *Valdez-Moreno et al., 2021*). The early life history of neotropical inland water fishes remains poorly documented but is of increasing interest due to the threats to these ecosystems and the vulnerability of eggs and larvae to environmental perturbations.

In this study, we surveyed the Chile Verde Lagoon, Quintana Roo, Mexico to establish a baseline species inventory of the fish fauna of the region, including early life stages identified using DNA barcodes.

## MATERIALS & METHODS

### Study area

The Yucatan Peninsula is a massive carbonate platform composed of limestone, dolomite, and evaporites with an extensive underground karstic system (*Perry, Velazquez-Oliman & Marin, 2002*). The age of the platform ranges from Cretaceous to Holocene, with the bedrock becoming younger toward the north (*Bauer-Gottwein et al., 2011*). Chile Verde is an oligohaline lagoon in the southeastern Yucatan Peninsula in Quintana Roo, Mexico (18.829°, −88.186°), 26 km by 0.8 km at its widest part, and a maximum of 3 m above sea
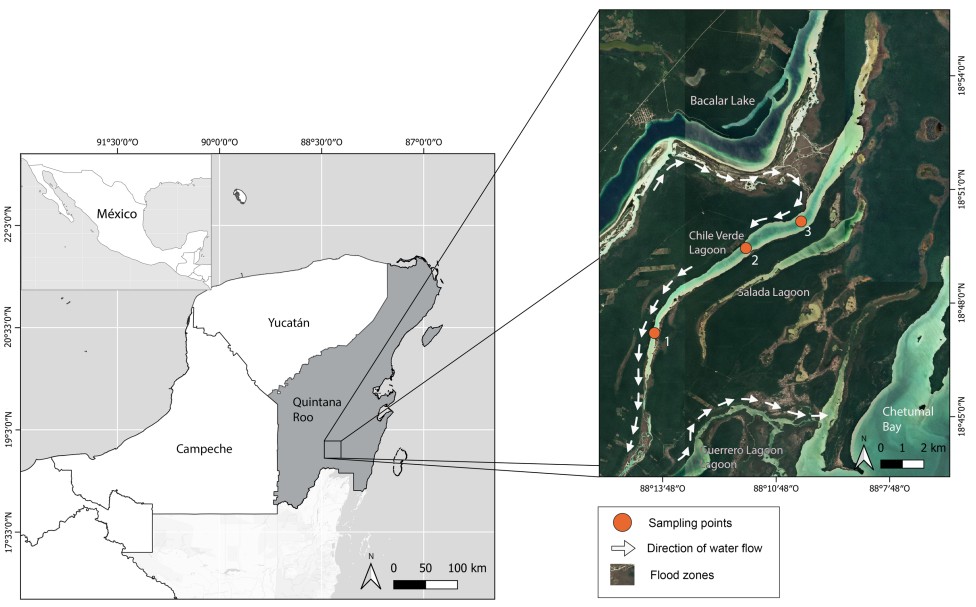

**Figure 1** Location of Chile Verde Lagoon in Yucatan Peninsula and sampling points.

level. It is mainly composed of highly permeable limestone, allowing water to filter into underground rivers. The lagoon is linked to several systems: to the west, it connects with Lake Bacalar, to the southeast with Salada Lagoon, Guerrero Lagoon, and Chetumal Bay, through intermittently flooded areas (Fig. 1). The prevailing climate is warm sub-humid with rainfall from May to October and an annual average of 1,200 mm (*Rosado-May, Romero-Mayo & Navarrete, 2002*).

The chemistry of the Chile Verde Lagoon itself has not been evaluated; however, nearby Lake Bacalar has elevated concentrations of carbonate and/or sulfate ions because of the dissolution of limestone bedrock in waters circulating in a connected karst system (*Gischler, Gibson & Oschmann, 2008*). The lagoon and associated waters are an althalassohaline system, defined as an inland lake of no marine origin with an ionic composition dominated by calcium, magnesium, and sulfate *vs.* sodium and chloride in the ocean (*Alcocer & Escobar, 1994*; *Cole, 1979*).

## Sampling

We collected fishes at three sites along the lagoon during the nights of the new moon between February and May 2022. Adults and juveniles were caught using seine nets, hand nets, and cast nets with a sampling effort of three people per two hours at each point. Also, three traps per site were left overnight.

Larvae were collected with light traps left overnight as well as with plankton net tows. Light traps (600 lumens) were placed at two locations per site (littoral and limnetic) at a depth of 1 m and operational from 18:00 to 07:00 hrs (*Montes-Ortiz & Elías-Gutiérrez, 2018*). Trawls used a plankton net with a mouth diameter of 50 cm and a mesh of 333 μm for 10 min at 2–3 knots. Samples obtained were filtered with a 50 μm sieve, fixed in

chilled ethanol 96% and stored for a week at −18 °C. A multiparameter Hanna HI98194 with a HI7698194 probe was used to measure temperature, conductivity, and pH at each sampling point. The collection permit was provided by Secretaría de Agricultura, Ganadería, Desarrollo Rural, Pesca y Alimentación (SAGARPA) (PPF/DGOPA-006/22). Each of 15 plankton-net samples and 28 light trap samples were analyzed for an estimate of biomass, calculated as wet weight from suctioned liquid (*Zavala-García & Flores-Coto, 1989*) using a 25 cm Hg vacuum pump and repeated three times per sample.

## Taxonomical identification

We identified adult and juvenile fishes following *Schmitter-Soto (1998)* and *Greenfield & Thomerson (1997)*. Larvae were counted and sorted to morphotypes based on meristic counts, shape, body size, eye shape, pigmentation, and notochord development (*Richards, 2006*). In the absence of taxonomic keys for inland water fishes larvae in this region, we used *Richards (2006)* and a few available larval descriptions (*Beeching & Pike, 2010*; *Uh-Navarrete et al., 2021*). The species names followed Eschmeyer's Catalog of Fishes (*Fricke, Eschmeyer & Van der Laan, 2023*).

## DNA Barcode analysis

One to five specimens of each morphotype were selected for DNA sequencing, specimens were stored as vouchers in the collection of fishes (ECO-CH-P) and ichthyoplankton (ECO-CH-LP) at El Colegio de la Frontera Sur. Each specimen was photographed under a Nikon SM2 745T stereomicroscope with an Eos Rebel T7i camera. We tissue-sampled the entire larva for those under 3 mm TL, between 3 and 4 mm TL we submitted the tail end, and in larger larvae a 1–2 mm$^3$ sample of muscle was cut away. Between each tissue collection, the forceps were sterilized with chlorine diluted in water (1:5) and rinsed with 96% ethanol.

DNA was extracted using the glass-fiber method (*Ivanova, Dewaard & Hebert, 2006*). The cytochrome oxidase I (COI) gene was amplified using FishF1 and FishR1 primers (*Ward et al., 2005*) and Zplank primers (*Prosser, Martínez-Arce & Elías-Gutiérrez, 2013*). The PCR mixtures contained a final volume of 12.5 μL, prepared with 6.25 μL of 10% D-(+)- trehalose dihydrate, 2 μL of Milli-Q water, 1.25 μL of 10x Platinum Taq buffer, 0.625 μL of 50 μM MgCl2, 0.0625 μL of 10 μM dNTP, 0.125 μL of each forward and reverse primer (10 μM), 0.06 μL of Platinum Taq and 2 μL of DNA template. The reactions for the Fish primers were cycled at 94° for 1 min, followed by 35 cycles at 94 °C for 30 s, 52 °C for 40 s and 72° for 1 min, with a final extension of 72 °C for 10 min. For the Zplank PCR process we followed *Prosser, Martínez-Arce & Elías-Gutiérrez (2013)*. PCR products were visualized on 2% agarose gels (E-Gel 96 Invitrogen) and extracts were sequenced by Eurofins Scientific in Louisville, Kentucky.

The sequences were edited using CodonCode v.3.0.1 and uploaded to the Barcode of Life Database (*BOLD Systems, 2023*, boldsystems.org) in the project "Freshwater fish from Chile Verde, Quintana Roo" (dx.doi.org/10.5883/DS-CHVER). All sequences are available on GenBank *via* accessions OR138669–OR138981. These were examined with BOLD tools for detecting ambiguous bases, stop codons or possible mistakes. Sequences that passed quality control were assigned to a BIN by the BOLD algorithm.

A BIN (Barcode Index Number) is a code for an algorithmically derived cluster of barcode sequences. The BIN assignment can assist in species identification, especially when the species IDs of the records in the BIN are supported by vouchers and photographs that are sufficient to identify fishes to the species level and exclude similar appearing species. The species suggested by the BIN is not necessarily the species in question, since, as *Robertson et al. (2022)* points out "there are many exceptions to the "*one-species, one-BIN*" *concept: either multiple BINs per species, indicating genetically divergent populations within species (usually allopatric, but not always), a subset of which are putative new cryptic species awaiting morphological confirmation; or shared BINs by two or more species that retain shared or closely related haplotypes due to a short time since speciation, to incomplete lineage sorting, or to a small degree of hybridization*". Each identification needs to be evaluated for alternative IDs, including by examining nearest neighbor BINs and BINs assigned to congeners. In this study, we verified the species assignment with our own voucher specimens, photographs, geographic information, and evaluation of other submitted identifications by other projects in BOLD.

## Data analysis

Our completed set of sequences were placed into a neighbor-joining tree (NJ) based on the Kimura two-parameter (K2P) distance model which graphically represents genetic distances between taxa using MEGA v7 software (*Kumar et al., 2018*). In barcode studies, this kind of trees are considered an "Id tree".

Species richness was calculated and graphed on an accumulation curve using EstimateS v9.10 software (*Colwell, 2013*). For the curve, the data were grouped into three samples (light trap, standard plankton net, and fishing gear for adult fishes) per field trip (February–May). The classic Chao 1 estimator was used with an extrapolation of 30 samples, 1000 randoms, and considering $\leq 5$ specimens captured as rare species. The Spearman rank-correlation coefficient (Rho) between biomass and abundance was calculated with a significance level of 0.5 and 42 degrees of freedom, using SPSS v29 software.

# RESULTS

## Species identification

We collected 197 adult and juvenile fish specimens, and 3,722 larvae. We visually assigned the larvae to 20 different morphotypes; most types were identifiable only to the family level for seven families (Belonidae, Cyprinodontidae, Gobiidae, Clupeidae, Cichlidae, Engraulidae, Hemiramphidae), three to genera (*Strongylura, Chriodorus, Microgobius*) and only one to species (*Chriodorus atherinoides*). Adults and juveniles were identified to 23 species (Table 1), with two, *Ophisternon aenigmaticum* and *Hypanus* sp., observed but not collected.

We sequenced 306 specimens of adults, juveniles, and larvae and obtained 296 COI sequences (96.7% success rate; 648 bp and one 581 bp). The sequences matched to 25 BINs, and we could identify all to species: *i.e.,* 25 species in 24 genera and 13 families (Fig. 2 and Table 1). The distribution of K2P intraspecific distances ranged from 0 to 1.57%, while interspecific distances ranged from 11.9% to 12.28%, fulfilling the "barcode

Table 1 **Species collected in Chile Verde lagoon.** The molecular data show the number of organisms sequenced and the assigned BINs. (#Seq) number of sequences obtained for each species. (+) species in the adult stage that were initially identified morphologically. (F) 01/February/22. (M) 02/March/22. (A) 02/April/22. (My1) 02/May/22. (My2) 31/May/22.

| Taxonomy | | | Months | | | | | Molecular data | |
|---|---|---|---|---|---|---|---|---|---|
| Order | Family | Species | F | M | A | My1 | My2 | #Seq | BIN |
| Batrachoidiformes | Batrachoididae | *Batrachoides gilberti* + | * | | | | | 1 | AAF9670 |
| Siluriformes | Heptapteridae | *Rhamdia guatemalensis* + | * | | | | * | 2 | ACF4952 |
| Characiformes | Characidae | *Astyanax bacalarensis* + | * | * | * | * | * | 8 | AAA6360 AEY6469 |
| Perciformes | Gerreidae | *Eugerres plumieri* + | | | * | | | 1 | AAB4701 |
| Beloniformes | Belonidae | *Strongylura notata* + | * | * | * | | | 5 | AAC4691 |
| | Hemiramphidae | *Chriodorus atherinoides* + | * | * | * | * | * | 27 | AAD0222 |
| Cyprinodontiformes | Cyprinodontidae | *Cyprinodon artifrons* + | * | * | * | * | * | 27 | AAA8182 |
| | | *Floridichthys polyommus* + | * | * | * | * | * | 25 | AAA6554 |
| | | *Jordanella pulchra* + | * | * | | * | * | 4 | AAD5728 |
| | Poeciliidae | *Gambusia yucatana* + | * | * | | * | | 5 | AAA4520 |
| | | *Gambusia sexradiata* | | * | | | | 2 | AAF5311 |
| | | *Poecilia mexicana* + | | | * | * | * | 5 | AAA4518 |
| | | *Belonesox belizanus* + | | | | * | | 1 | AAD3459 |
| Clupeiformes | Engraulidae | *Anchovia clupeoides* + | * | | | | * | 8 | ACV0719 |
| | Clupeidae | *Dorosoma petenense* + | * | * | * | * | * | 38 | AAC3463 |
| Cichliformes | Cichlidae | *Vieja melanurus* + | * | * | * | * | | 14 | AAB9907 |
| | | *Thorichthys meeki* + | * | * | | * | | 13 | AAA4760 |
| | | *Trichromis salvini* + | | | * | * | * | 4 | AAD0719 |
| | | *Petenia splendida* + | | | | * | * | 4 | AAD6009 |
| | | *Mayaheros urophthalmus* + | * | * | | | | 2 | AAB5118 |
| Gobiiformes | Eleotridae | *Gobiomorus dormitor* + | | | * | | | 1 | AAB7696 |
| | Gobiidae | *Lophogobius cyprinoides* + | * | * | * | * | * | 50 | AAB6671 |
| | | *Microgobius microlepis* | * | * | * | * | * | 33 | AAX4201 |
| | | *Bathygobius soporator* | | | | * | | 1 | AAA7195 |
| | | *Gobiosoma yucatanum* | * | * | * | * | * | 15 | ACV0831 |
| Myliobatiformes | Dasyatidae | *Hypanus* sp. | * | | | | | – | - |
| Synbranchiformes | Synbranchidae | *Ophisternon aenigmaticum* | | | | * | * | – | - |

**Notes.**
*presence of the species in each month.

gap" promoted by *Hebert et al. (2003)*. *Astyanax bacalarensis* had the greatest intraspecific distance (1.57%), the remainder were ≤ 1.09%.

Our inventory documented the presence of 27 fish species in the Chile Verde Lagoon (Table 1). The estimated accumulation curve shows a rapid increase in species during the first sampling with a slow tendency to asymptote (Fig. 3). Following the Chao 1 estimator, 36 fish species are expected in the Chile Verde Lagoon, hence the number of species found in this study composed 75% of the estimated potential richness of the system using that model.

The family Cichlidae was the most diverse within the lagoon, with five species (*Mayaheros urophthalmus*, *Vieja melanurus*, *Thorichthys meeki*, *Petenia splendida*, and *Trichromis*

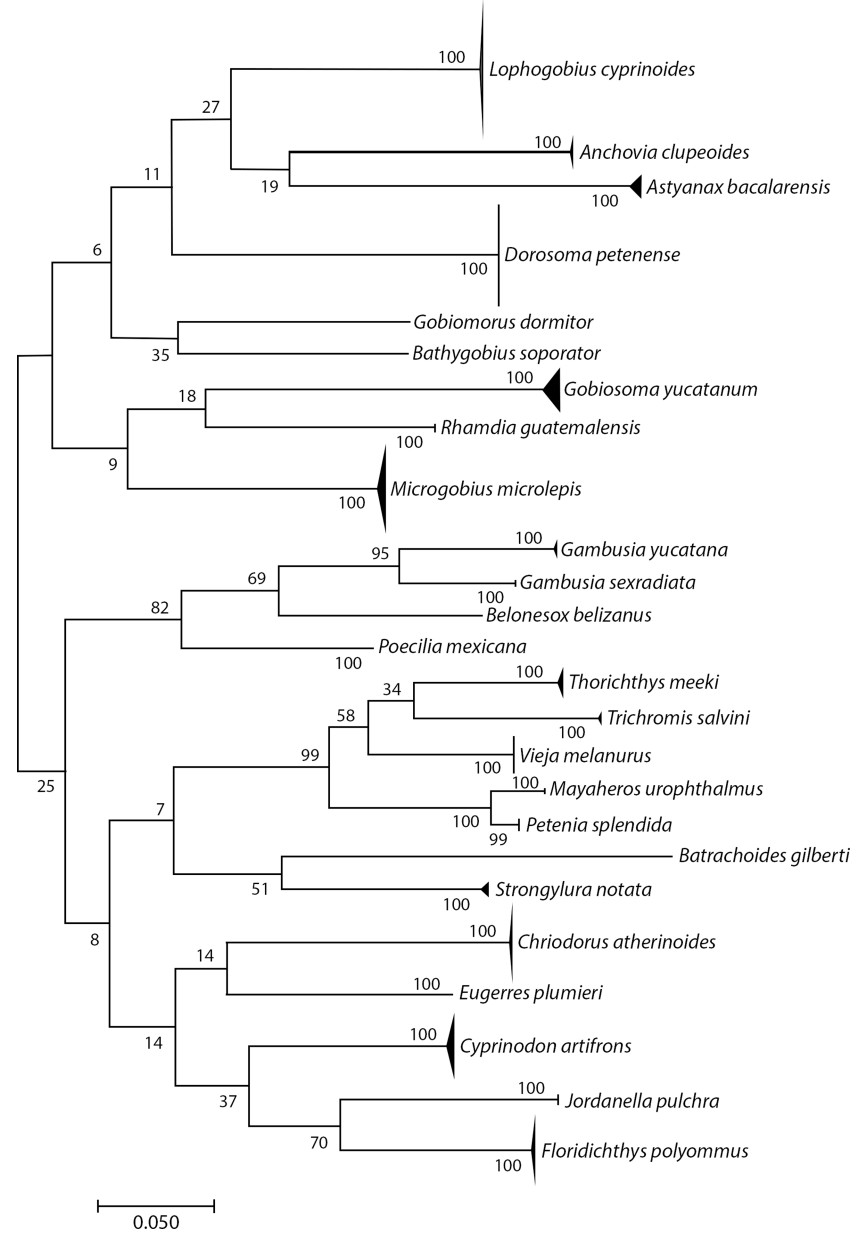

**Figure 2** Simplified neighbor-joining tree showing the species identified using the COI gene.

*salvini*); all collected as adults and also four larvae identified as *Petenia splendida*. Four species of the family Gobiidae were collected as larvae (*Lophogobius cyprinoides, Microgobius microlepis, Bathygobius soporator,* and *Gobiosoma yucatanum*), and as adults for *Lophogobius cyprinoides*. The family Poeciliidae included four species (*Gambusia yucatana, Gambusia sexradiata, Belonesox belizanus,* and *Poecilia mexicana*), all collected as adults. The remaining families had between one and three species in differing stages of development (Tables 1 and 2).
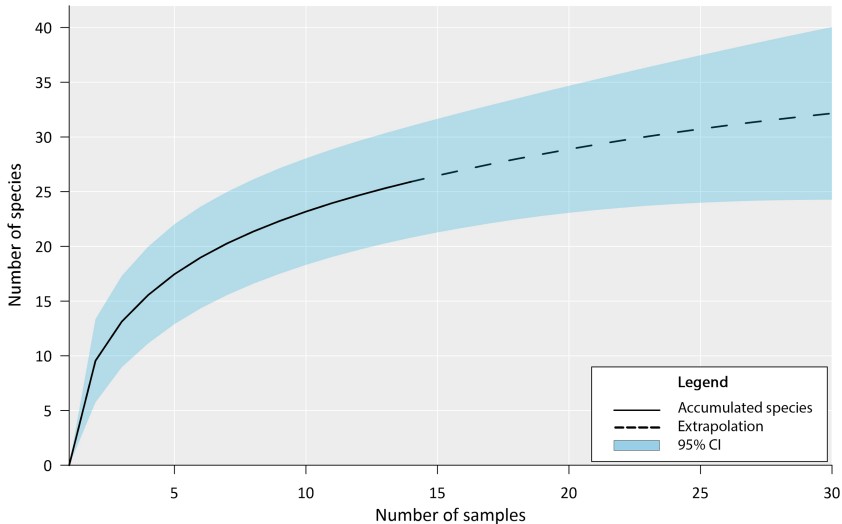

**Figure 3** Fish species richness from four months of sampling in Chile Verde Lagoon (February to May 2022), with extrapolation to 30 samples. For the construction of the accumulation curve, data were grouped into light trap samples, standard plankton net samples, and adult sampling gear for five sampling events.

## Sampling gear and abundance

The efficiency of the collecting method varied for adults. The most efficient gears were hand nets and cast nets employed at night, which collected 18 taxa (66.6%). Hand nets proved to be especially effective overnight, while fishes are less mobile, whether on the bottom (*V. melanurus, T. meeki, T. salvini, M. urophthalmus* and *L. cyprinoides*) or on the surface (*Strongylura notata, C. atherinoides*, and *G. yucatana*). Traps for adults captured only six species in low numbers; nevertheless, three species were only collected in traps, *i.e., Batrachoides gilberti, Gobiomorus dormitor,* and *Rhamdia guatemalensis.*

The ichthyoplankton sampling also varied with gear both in composition and abundance (Fig. 4 and Table 2). The standard plankton net yielded the largest catch (1,789 specimens), dominated by *M. microlepis, L. cyprinoides*, and *Dorosoma petenense*. The latter two species were captured in the yolk-sac stage, indicating a recent hatch. Limnetic light traps had less abundance than plankton nets, yielding 1,203 specimens; the catches were composed of similar species and dominated by the same taxa. In contrast, littoral light traps had the lowest abundance, capturing 730 specimens, and dominated by *L. cyprinoides, Cyprinodon artifrons* and *Floridichthys polyommus*. However, two species, *P. splendida* and *Jordanella pulchra*, were only collected with this gear.

The number of fishes collected varied widely between species, with clupeids and gobiids most abundant, accounting for 87.9% of individuals (Table 2). The clupeid, *D. petenense*, was the most abundant species, with 1,260 larvae and one adult collected. Of the gobies collected, *L. cyprinoides* (1,153 specimens) and *M. microlepis* (837 specimens) accounted for most of the catch. The latter was only collected as larvae, up to the settlement stage (stage prior to juvenile) (Fig. 5). Several other species were represented by one or two specimens,

**Table 2 Abundances and sampling gear used to collect each species.**

| Specie | Adult and juvenile | Settlement and larvae | Fishing gear |
|---|---|---|---|
| *Batrachoides gilberti*[a] | 1 | – | Trap |
| *Rhamdia guatemalensis*[b] | 2 | – | Trap |
| *Astyanax bacalarensis*[b] | 23 | – | Trap and seine net |
| *Eugerres plumieri*[a] | 1 | – | Cast net |
| *Strongylura notata*[a] | 2 | 4 | Hand net, cast net, light trap, and standard plankton net |
| *Chriodorus atherinoides*[a] | 16 | 17 | Hand net, light trap, and standard plankton net |
| *Cyprinodon artifrons*[a] | 30 | 180 | Cast net, seine net, light trap, and standard plankton net |
| *Floridichthys polyommus*[a] | 20 | 156 | Cast net, seine net, light trap, and standard plankton net |
| *Jordanella pulchra*[a] | 3 | 1 | Light trap |
| *Gambusia yucatana*[a] | 15 | – | Hand net and seine net |
| *Gambusia sexradiata*[b] | 21 | – | Seine net |
| *Poecilia mexicana*[a] | 7 | – | Hand net and seine net |
| *Belonesox belizanus*[a] | 1 | – | Cast net |
| *Anchovia clupeoides*[a] | 6 | 2 | Light trap |
| *Dorosoma petenense*[a] | 1 | 1260 | Light trap and standard plankton net |
| *Vieja melanurus*[a] | 13 | – | Cast net and hand net |
| *Thorichthys meeki*[b] | 16 | – | Cast net and hand net |
| *Trichromis salvini*[b] | 9 | – | Trap, cast net, and hand net |
| *Petenia splendida*[b] | 1 | 8 | Trap, cast net and light trap |
| *Mayaheros urophthalmus*[a] | 2 | – | Cast net and hand net |
| *Gobiomorus dormitor*[a] | 1 | – | Trap |
| *Lophogobius cyprinoides*[a] | 6 | 1153 | Hand net, light trap, and standard plankton net |
| *Microgobius microlepis*[c] | – | 837 | Light trap and standard plankton net |
| *Bathygobius soporator*[a] | – | 1 | Light trap |
| *Gobiosoma yucatanum*[a] | – | 14 | Light trap and standard plankton net |
| *Hypanus* sp.[a] | 1 | – | Observation |
| *Ophisternon aenigmaticum*[b] | 4 | – | Observation |
| Indeterminate | – | 89 | – |
| **Total** | **197** | **3722** | |

**Notes.**
[a] euryhaline species.
[b] freshwater species.
[c] marine species.

including *Batrachoides gilberti, Eugerres plumieri, B. belizanus, G. dormitor, B. soporator, Hypanus* sp., and *R. guatemalensis.*

## Ichthyoplankton biomass

Three families accounted for the most biomass: the Gobiidae, Clupeidae, and Cyprinodontidae. The first two were also the most abundant (Fig. 6A), the correlation between biomass and abundance was 0.402 ($\alpha = 0.05$; $P = 0.008$). The total biomass of the entire ichthyoplankton catch was 15.77 g, with an average of 0.43 g among the samples. Biomass was similar for most of the months sampled, the lowest biomass occurred in February and April (2.58 and 2.61 g respectively) up to 4.68 g in May (Fig. 6B).
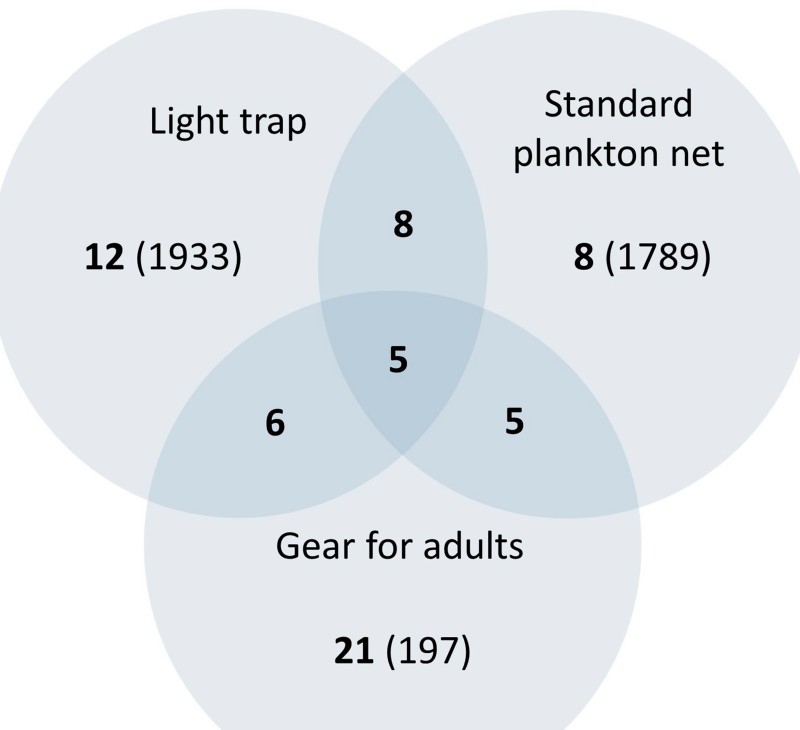

**Figure 4  Venn diagram comparing the richness collected by art sampling.** The number in bold indicates the species collected, the number in brackets is the abundance, and the number in the overlapping area indicates shared species. It should be noticed that littoral and limnetic light traps are together.

## Water parameters

Here we presented the first punctual data on some water parameters of the Chile Verde Lagoon.

Our physical survey revealed an overall mean depth of 2.14 ± 1.15 m, with a shallow southern zone of 0.9 ± 0.05 m, a central zone of 3.2 ± 0.51 m, and a northern zone of 2.7 ± 0.33 m. The pH was consistent at most points with a mean of 8.3 ± 0.58 and an average conductivity of 4371 ± 379 µS/cm, equivalent to a salinity of 2.2 ± 0.16 PSU. Water temperature in February averaged 24.05 ± 0.75 °C and in March 24.87 ± 1.80, with the subsequent months about 4 °C warmer, and consistent ($\bar{x} = 28.90 \pm 0.55$ °C). This change of temperature is less noticeable in deeper systems, such as the nearby Lake Bacalar with a mean depth of 13.3 m (data taken from *De Jesús-Navarrete & Legorreta, 2022*) and an annual variation of approximately 2 °C (*Tobón Velázquez et al. 2019*).

## DISCUSSION

### Species identification

Morphological identification was effective for adults and juveniles, we identified all we captured but one, the exception being *Gambusia sexradiata* because the fish was lighter
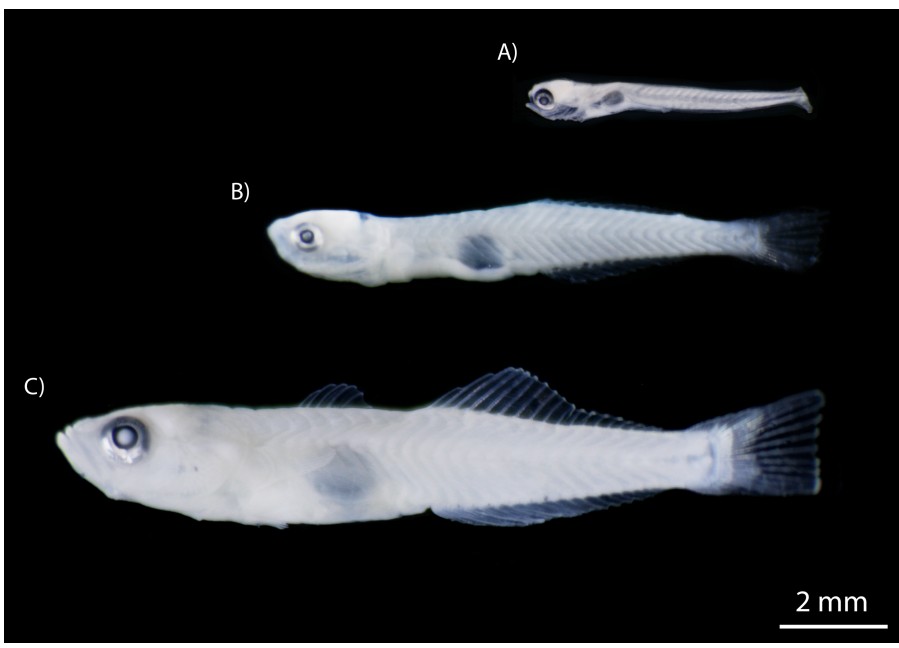

**Figure 5** **Developmental stages of *M. microlepis* larvae captured in the Chile Verde Lagoon.** (A) Larvae in preflexion; no fin rays. (B) Larvae in flexion; 1st dorsal fin rays incipient, 2nd dorsal and anal fin rays developed. (C) Settlement stage; fully developed fin rays.

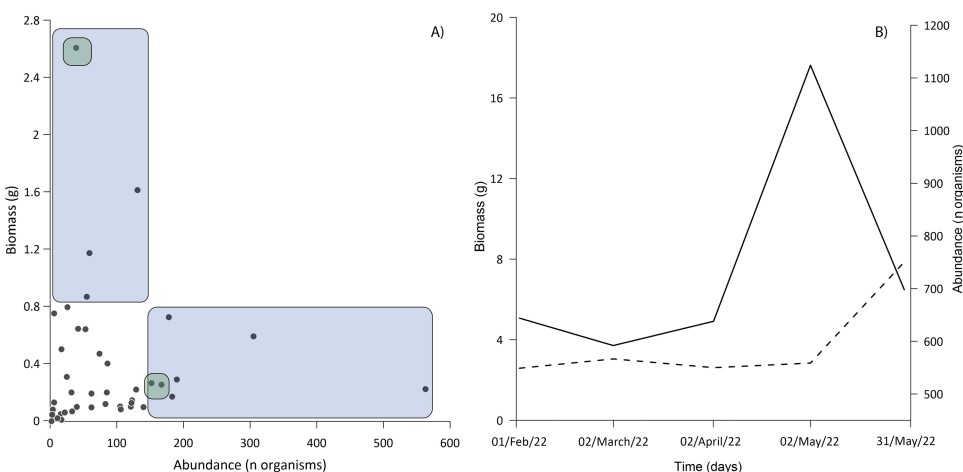

**Figure 6** **Relationship between abundance and biomass.** (A) The blue area shows samples dominated exclusively by Gobiids and Clupeids; the intersection of green areas shows samples dominated by Gobiids, Clupeids and Cyprinodontids. (B) Behavior of ichthyoplankton biomass (dashed line) and abundance (solid line) during the sampling period. All standard deviations of ichthyoplankton biomass from each sampling event were <0.02%.

than usual, and we could not interpret the spot under the eye that distinguishes species (*Greenfield & Thomerson 1997*). Because of the lack of ID keys for larvae, we were only able to identify one larval type to species, *i.e., Chriodorus atherinoides*. In contrast, DNA barcodes identified all stages of all species sequenced. Similar findings have been reported for other tropical freshwater fishes surveyed in the same manner (*Frantine-Silva et al., 2015*; *Panprommin et al., 2020*).

*Astyanax bacalarensis* had unusual results from sequencing, it had high divergence in COI sequences and occupied two separates, but close, BINs (AAA6360 and AEY6469). This species has been described recently by *Schmitter-Soto (2017)*, separating it from *A. aeneus, A. altior,* and *A. angustifrons*. After our review of the specimens of Chile Verde, we noted some of them had 12 or 13 pectoral-fin rays *vs.* 10 or 11 pectoral-fin rays for the type population. This suggests there may be more than one species with nominal *A. bacalarensis*, a good example of how DNA barcode surveys can reveal cryptic speciation. The sequences of two of our specimens matched to a very large BIN AEY6469 that included 11 different identifications (in five genera) by various contributors with specimens collected from the USA to Brazil. The significance of this extreme genetic heterogeneity in some characid fishes remains to be elucidated.

While most species in this study were found to conform to the one species/one BIN paradigm, *Cyprinodon artifrons* shared the same BIN AAA8182 with seven other congeners from Chichancanab lake. This shared BIN is likely the result of a recent radiation of young endemic species of *Cyprinodon* species that have developed morphological differences with minimal genetic divergence (*Strecker, 2006*) similar to adaptive radiations of cichlids in the Lakes of Africa (*Seehausen, 2015*).

## Inventory of species

Of the 27 species we identified, all but *Microgobius microlepis,* have been previously recorded in inland waters from this region of Mexico (*Schmitter-Soto, 1998*; *Miller, Minckley & Norris, 2009*; *Schmitter-Soto et al., 2009*). Records of this species in Mexico are limited; *Birdsong (1981)* reports collections from northeastern Yucatan peninsula between 1960 and 1980 (*GBIF, 2022*). In 2000, two specimens assigned only to the genus *Microgobius* were collected in Ascension Bay (*Vásquez-Yeomans, 2000*), the same author also found one specimen from Isla Contoy (unpublished data). In contrast, we document for the first time the occurrence of this presumed marine species in an oligohaline system (athalassohaline) and notably in large numbers (837 specimens). Their presence in the Chile Verde Lagoon may result from their preference for shallow water and a substrate of calcareous sediments (*Birdsong, 1981*; *Tornabene, Van Tassell & Robertson, 2012*). All specimens of this species collected between February and May were larvae or at the settlement stage (Fig. 5). However, eight months after our last sampling, we observed adult specimens in their burrows, and collected four males and two pregnant females (Anexo 1), confirming that the species likely maintains an established resident population in the lagoon. *Powell et al. (2007)* reported that in Florida Bay, peak numbers of settled juveniles were trawled on the bottom in November, indicating settlement in late summer with spawning some time before that.

Our results show that the population in the Chile Verde Lagoon have a broad reproductive period that includes the northern winter months.

Some species are economically important locally, such as *Eugerres plumieri, Mayaheros urophthalmus,* and *Petenia splendida,* which are fished by local fishermen, the latter two species have some potential for aquaculture. *Cyprinodon artifrons* was found in large numbers, both as larvae and adults, and is also common in Chetumal Bay (*Schmitter-Soto et al., 2009*; *Schmitter-Soto & Herrera-Pavón, 2019*). Their larvae were recently recorded in Lake Bacalar (*Elías-Gutiérrez et al., 2018*; *Uh-Navarrete et al., 2021*), indicating that the Chile Verde Lagoon is a connection between Chetumal Bay and Lake Bacalar.

The high diversity in the tropics with frequent rare species requires large samples to accurately estimate species richness (*Chao et al., 2009*; *Gotelli & Colwell, 2011*). Our estimate of 75% is likely low, since we obtained many singletons and doubletons and did not exhaustively sample all potential microhabitats. It is important to note that most of the species we found are known to be euryhaline, while seven are considered freshwater and *M. microlepis* is generally considered to be a marine species (Table 2) (*Froese & Pauly, 2021*).

### Sampling gear and abundance

In larval sampling, two of the most abundant species (*D. petenense, L. cyprinoides*) were fishes known to also occur in Lake Bacalar and the Chetumal Bay (*Montes-Ortiz & Elías-Gutiérrez, 2018*; *Schmitter-Soto & Herrera-Pavón, 2019*). Unexpectedly, larvae of the typically marine *M. microlepis* were also one of the more abundant species. Larval numbers peaked in May, driven by a pulse of breeding of *L. cyprinoides* (Fig. 6B), which corresponds to the reproductive season reported by *Schmitter-Soto (1998)*. The presence of large numbers of larvae indicates the lagoon may be an important nursery area for fishes within the system. The low correlation between biomass and abundance is explained by the selectivity of gear for small larvae, disconnecting abundance and biomass values. Our results demonstrate the need to use a variety of sampling gear to fully inventory any tropical inland water system (*Vásquez-Yeomans et al., 2011*; *López-Vila et al., 2014*). All methods are selective to varying degrees, such as light traps that only capture species with positive phototaxis (*Mueller & Neuhauss, 2010*; *Massure et al., 2015*).

## CONCLUSIONS

We recorded 27 taxa for Chile Verde Lagoon, this result was mainly due to the use of a variety of sampling gear targeting various life stages.

The low success rate in the morphological identification of larval stages was mainly due to the lack of descriptions and few diacritical characteristics that make it difficult to identify them at species. This is why we highly recommend using more of an identification method.

In contrast, DNA barcodes identified all species collected. This proved that it is an effective tool for identifying tropical fishes in all stages of development.

The high biomass and abundance of larvae of the clupeids, gobiids, and cyprinodontids indicate that the Chile Verde Lagoon could be an important reproduction site for these groups.

*M. microlepis* is reported for the first time in an oligohaline lagoon. The presence of them in larval, juvenile, and adult stages in different months confirms that the species likely maintains a residence population in Chile Verde Lagoon.

Finally, this is the first step to discovering the fish fauna of the Chile Verde Lagoon. The information obtained from this study provides us with essential knowledge for future environmental assessments, system biomonitoring, and fish fauna conservation in a region where the impact of tourism is growing continuously without control.

## ACKNOWLEDGEMENTS

To Benjamin Victor for his valuable contribution to the English revision and his comments. To Manuel Elías Gutierrez for his unconditional field help and for his accurate comments. Alma Estrella García Morales from the Mexican Barcode of Life (MEXBOL) node Chetumal assisted with DNA extraction, PCR reactions, and sequence edition of all materials presented here. To José Ángel Cohuo Colli and Elda Aurora Canul Ramírez for assisted us during field trips. To Humberto Bahena Basave and Jonathan Jesús Reyes Poot for gave us support with photographs. All the members of the ejido ''La Península'' kindly allowed us access to the lagoon and gave us the facilities to carry out this project. Sara LeCroy and Michael J. Andres of the University of Southern Mississippi for provided us with the original field data sheet and catalog number for the first records of *M. microlepis* in Mexico.

### Funding

This study was financed by Consejo Nacional de Ciencia y Tecnología (CONACYT), scholarship number 1020183, awarded for pursuing a Master of Science in Natural Resource Management and Rural Development. The funders had no role in study design, data collection and analysis, decision to publish, or preparation of the manuscript.

### Grant Disclosures

The following grant information was disclosed by the authors:
Consejo Nacional de Ciencia y Tecnología (CONACYT), scholarship number 1020183, awarded for pursuing a Master of Science in Natural Resource Management and Rural Development.

### Competing Interests

The authors declare there are no competing interests.

### Author Contributions

- Adrian Emmanuel Uh-Navarrete conceived and designed the experiments, performed the experiments, analyzed the data, prepared figures and/or tables, authored or reviewed drafts of the article, request for collection permits, and approved the final draft.
- Martha Valdez-Moreno conceived and designed the experiments, analyzed the data, authored or reviewed drafts of the article, request for collection permits, and approved the final draft.

- Mariana E. Callejas-Jiménez analyzed the data, prepared figures and/or tables, authored or reviewed drafts of the article, and approved the final draft.
- Lourdes Vásquez-Yeomans analyzed the data, authored or reviewed drafts of the article, and approved the final draft.

### Animal Ethics

The following information was supplied relating to ethical approvals (i.e., approving body and any reference numbers):

Secretaria de Acuícultura, Ganadería, Desarrollo Rural y Pesca

### DNA Deposition

The following information was supplied regarding the deposition of DNA sequences:

The sequences are available in the Supplementary File.

### Data Availability

The sequences are available in the Supplementary File and GenBank: OR138669–OR138981.

### Supplemental Information

Supplemental information for this article can be found online at http://dx.doi.org/10.7717/peerj.16285#supplemental-information.

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
