# Peer review of "Discovering the fish fauna of a lagoon from the southeast of the Yucatan Peninsula, Mexico, using DNA barcodes"

_PeerJ, doi:10.7717/peerj.16285_

## Round 0.1 · original submission · Major Revisions

The reviews varied a lot, but each of them made very useful points that you need to address when revising this manuscript, even if you decide not to resubmit to PeerJ. Despite the fact that reviewer 2 recommended rejection I think it would be better for you to invest the effort in a major revision that deals with substantial issues raised by all three reviewers.

My comments:

A lot of work is needed to bring the English writing in the manuscript up to an acceptable standard. That is the authors' responsibility as PeerJ does not provide such editing as a standard service. Running all the text through Grammarly. should help…this is a free program that helps correct grammar. If you can also get a fluent English speaker to read the updated manuscript. Another free program that I find very useful for translations is DeepL, which translates sections of text from Spanish to English and vice versa. However, it is not perfect and the results always need careful inspection. That combination alone would provide a much better manuscript in terms of the use of English.

Names of fishes. There is no indication of a source of names you used. Catalog of Fishes is one of the best (FishBase is not) and it shows that have been some changes. E.g. it is now Jordanella pulchra not Garmanella pulchra. You should check all names against such a source and indicate the source.

Chile Verde lagoon is not freshwater. Your results state that its salinity is about 2.2 psu. That’s considerably higher than freshwater, which is <0.5 psu and is often around 0.1. So your study area is an oligohaline lagoon (<0.5-5.0 psu) rather than a freshwater lagoon. Figure 1 does not show a connection between that lagoon and Chetumal Bay that is visible on Google Earth. This provides a connection of Chile Verde to the sea and explains why it is oligohaline. Redraw that figure to show the connections and include text about the oligohaline nature of this lagoon. Do not refer to this lagoon as freshwater and do reorient all results and discussion about the fact that it is oligohaline. The fact that over half the fish species routinely occur in marine to brackish water environments is consistent with an oligohaline environment. Finally, correct the latitude/longitude markings on that figure. They are not accurate or complete.

All authors are responsible for the content of each one of their papers and need to carefully review the results and discussion and use of the English language!

Reviewer 1. Had useful comments about the organization of barcode data. and corrections to the English. Yes.. It will be Opsanus dichrostomus in your area, not O beta. I agree pretty much with everything that the reviewer indicated.

Reviewer 2 raised two main issues.

1. That too much of the manuscript is spent on information about fishing gear effectiveness and fish biomass, neither of which is really relevant to a barcoding study. I agree. I think move most of this material, including its references to a supplementary file, and provide a brief summary in the body of the paper, linked to the supplementary file.

2. That there is a problem with your analysis of the genetic data…that you need to recheck realignment of your DNA sequences. Since I am not sufficiently experienced with DNA analysis I asked a suitably experienced person if they thought Reviewer 2s comment is correct (simply by showing them the comment, without revealing the name of the reviewer or details about this manuscript). The response of that person was: “Yes, they definitely need to check their data as COI has a very uniform length. The reviewer is correct in that it is a coding gene & should not have any stop codons or indels so there shouldn't be any gaps. Especially gaps that are only 1 or 2 bp - if this is the case the chromatograms definitely need to be checked” So this is something substantial you must attend to in order to produce accurate barcode data before resubmitting the manuscript.

Reviewer 3. I agree with that reviewer that this study needs to be put in a wider context. There are lots of useful detailed comments.

Reviewer 1 ·

Basic reporting

issues with English, see additional below
excessive literature citations: I think there are too many literature citations, for example, four citations for biomass estimates is excessive (line 372)

Experimental design

clarify techniques as below

Validity of the findings

OK

Additional comments

This ms documents the inventory of fishes in a particular freshwater lagoon using DNA barcoding as the method of indentification. It is valuable both as a first example of using the technique for coastal freshwater systems that have not been surveyed in this manner, as well as documentation of a particularly interesting lagoonal area in Central America.

The science is well done and my main comments are in the style and wording of the presentation. It is often non-standard English and many phrases are apparently verbatim translations of Spanish wording, it needs some rephrasing, I offer a variety of corrections enumerated below.

It is important to state in methodology how a particular BIN lineage is identified- merely documenting a BIN number is not a species ID- each BIN needs to be evaluated, by vouchers, photographs, and who identified it etc. In this study, there is a lot of autovalidation, which is MOSTLY fine since the authors have collected vouchers and detailed voucher photographs- that needs to be discussed and emphasized. I do not doubt their IDs, I just believe there needs to be a standardization of how BINs are identified beyond saying that this BIN is Lophogobius etc.

However, one case is a clear demonstration of the issue- their ID of Opsanus beta (NOTE the BIN for Opsanus beta in Table 1 is a typo- the one listed is for an anchovy) actually has 3 different IDs in BOLD (AAD3683): O. beta, O. dichrostomus and O. phobetron. Which is it? note that the nearest neighbour BIN, 4.8% different (!), is also labeled as O. beta. On close examination, the other BIN AAD3684 is Opsanus beta, type location Texas, with specimens from Texas and the Gulf of Mexico. The specimens found in Yucatan must be Opsanus dichrostomus. I strongly recommend the authors write an additional paragraph discussing this as an example of caution in assigning identifications based on IDs in BOLD, by majority rules. Only after a taxonomic review of the details of the appearance, location, number of specimens, and related BINs with a close look at taxonomic questions for the taxon etc. can a species be identified. The authors should cite the FAO guide on Opsanus species ranges.

In Table 1, "percent ID" is not relevant and should be removed, it is auto-generated, since most of the records in most BINs is from the authors. In addition, percent ID is not a way to identify a BIN, it is not an assessment of the accuracy of the ID.

I may have missed it, but the GenBank numbers need to be mentioned somewhere.

Need to emphasize in the text and caption that the NJ tree in Fig. 2 is for illustrative purposes only, to graphically illustrate the intraspecific consistency vs. interspecific distances... only.... The making of trees implies phylogenetics and relatedness, and that does not belong in an identification study. The statistics of the robustness of the branching is misplaced, it assesses a phylogenetic question of reliability of branches and relatedness, and should be removed. And the 500 replicates is misplaced and should be deleted, that is all for phylogenetics. Whether two random species in a community are 18% different in sequence is meaningless. Virtually all the deep branches in the NJ tree are not robust and would change with a different selection. But the tree can remain since it shows that multiple sequences from the same species are almost the same.

detailed comments

use "fishes" for multiple species; round all percentages to one decimal point; instead of using capitalized families when an adjective is needed, just make the family into an adjective, i.e. gobiid or poeciliid; "settlement stage" is newly added midway on the paper, is it a larva? if so, make it "larvae at settlement stage", or define them as newly settled juveniles, if they are already transformed,

Remove the word epicontinental- it is jargon and not what the authors intend.


80- first sentence- no "The"; remove never; enumerate or include, not "consider"; more taxa than what??

consider is used three times in three lines- but consider means more to think about than to count for an inventory- "include" and its synonyms are better.

91- independent; delete "in which they are found"

92 effectiveness of

92-95- the citations are only a few examples, the wording implies they are universal - add "for example"

93- what does continent have to do with this? Hubert did a lot more

95- not needed "guarantee" or unequivocal= when meristics are diagnostic, larvae are easily identified, if there is only one species of a family in a certain ocean, then the identification to family is unequivocal. AND DNA barcodes are not guarantees, there may be close relatives that share DNA sequences- it is not rare.

101- these species implies you listed some species.. not "communities"- "and documented extinctions"?

102- the first time is unnecessary-- if one picks a small enough location, everything is first time-- better to say in this study we document the richness of...

109- no semicolon; New Moon not capitalized; "a total" unneeded, just five; at night; larval is the adjective; no capital after colon; no need for saying "two types of gear" and then 1) and 2)- just mention the two methods;

117 diameter of net?

126- replication details are only relevant for statistical comparisons- if purely descriptive don't need these details.

132- adult; fishes;

135 ray

139 fishes

150 %

151 no afterwards, every step in methods is "afterwards"

152- COI is not capitalized, no "the"

154- no "the"

165- just say that sequences that passed BOLD QC were assigned to BINs

183- just say "Fish larvae were.."

184- emphasize that they were identified only to the family level.. genus level.. (Table 1)

186- 21 species were identified from samples of adults and juveniles (Table 1)

187 no need to tell us what data, just (Table 1) suffices

191- no need for total, no need for "positive" (what is a negative sequence?), just say "Of the 306... 296 COI sequences..."

193- just say "with one 581 bp"- no need for last sentence, those would be removed in QC

195- this long sentence is ungrammatical and confused- need to say you used BIN assignment as the method to identify species, AND need to discuss how a BIN was confirmed as a species ID, most have multiple listed species names, often incorrect, so how did you evaluate the identification of a BIN- vouchers? majority rules? how many were the same as your morphological identification?

25 species... were assigned to separate BINs, --no need for requirements and quality, that is all BOLD QC, no need for "coincided", it is covered by "assigned to separate BINs".

198- all that matters are minimum and maximum intraspecific and interspecific distances, the average is not useful

202- no need for "total"; visual? this is the first time anything beyond DNA is mentioned?- put in parentheses (plus two species added based only on visual identifications)

ungrammatical to say corroborate a barcode associated with a BIN - 25 identified by BIN assignment??

205 calculating has no object, who is calculating?

206 I don't know what "reached" is

209 just say "collected as adults"

210 "and also 4 larvae identified for Petenia.."; "Four gobiids were identified from larval stages (...), with adults of Lophogobius also identified"

213- first time used, and undefined, "settlement stage", be consistent and use only larval, juvenile, and adult.

214 Four poecillid species were identified, all as adults (...).

229 in contrast, not unlike this

231 fishes, not organisms

232 "occupying % of the total abundances" is awkward, just "(87.9% of individuals)"

233 A single species of clupeid was collected, D. petenensis, but it was the most abundant species, with 1260 larvae (and one adult)

234 Two species accounted for almost all gobies collected, L. ..."


272- the success rate is irrelevant here, it is a technique attribute- important is that "all 25 species were identified by a BIN assignment".

276- The maximum intraspecific distance

277- the 2% threshold is obsolete, it needs to be eliminated from discussions.


294 "northwest of" means into the ocean, "in northeastern Yucatan"; 60s abbreviation is informal, say between 1960 and 1980

298 delete strictly, other Microgobius are euryhaline

299 "in large numbers"

302 sampling; demersal is not the reason, say "adults occupy burrows"

303- reproduction may be months earlier if a long PLD, say recent settlement event

304 remove the

306/7 ungrammatical sentence, delete it, not needed-- exhaustive implies huge effort,sampling doesn't "explain behavior", no mention before about seasonality, I had to go back to check which months the study included.

312- in large numbers, both as larvae and adults

321 is generally considered a marine species

323- delete this sentence, it is just a model, obviously "a year" is meaningless to go from 27 to 36 species.

324, acceptability of representativeness is not meaningful, who decides acceptable, and complex is relative, FW is much less complex than coral reefs-- better to say "The model predicts a higher diversity, however the number of samples to reach the predicted asymptote is large and the applicability to this system is unknown".

327- say "complete an inventory"

328- you did sample "specific habitats"- a wider range of microhabitats or..."

339 gear

340 collected a similar

343 gear

344- remove epicontinental, wrong word

348 first

352 were the most abundant larval type

354 corresponds

376 a shallow system is a vague terminology, applies to all shores

379 just deeper, not associated

382 presented high values is a long way of saying "higher than in equivalent freshwater systems"

383 values are not "given", say "Rather than saltwater attributes, these measures are accounted for by dissolved calcium carbonates..",

really that affects "salinity'"??

384- other studies

Reviewer 2 ·

Basic reporting

The paper # 81697 entitled: “Discovering the richness of fishes in a lagoon from
the southeast of the Yucatan Peninsula, using DNA barcodes” deals with the use of genetic markers (mitochondrial COI gene) in order to accurately identify fish species in a lagoon in the Yucatan Peninsula in southern Mexico. Whereas the manuscript address well the effectiveness of the use of barcodes to identify fish species in the mentioned system, I think the study makes a clear point of usefulness of the techniques specially to identify fishes in early developmental stages (eggs and larvae). In general the manuscript is well written, however there are places where the manuscript will benefit with some English proof reading. Some comments are given below and in a pdf version of this review. Besides the English issues, that are actually minor. I found a series of major issues, that I will point out below:
Whereas the MS makes a good point in use of barcodes, the MS on the other hand get loss in details of fish biomass and fishing gear effectiveness. Important topics no doubt, but they do not contribute to much or anything to main purpose of this study, that is to demonstrate the usefulness of barcodes a tool for fish identification. In both cases, but specially the effectiveness of fishing gear there is a huge body of literature that address this issue in a more complete form. The biomass data, is not contributing either to the barcoding part of the study, so, it becomes redundant. Besides, I think figure 6, where the main biomass results are presented, the variables are upside down (see comment below).

A second major issue appears after reviewing the supportive material and inspect the COI alignment. After a very quick inspection of the alignment, some gaps where notice it. It is important to point out to the authors that the COI is a protein-coding gene, so it should no include any gaps. I included here a stamen from Buhay 2009 “COI is a protein-coding gene, and as such, has an open reading frame. Open reading frames do not include stop codons or indels leading to gaps in the alignment which disrupt the translation of the DNA sequence into amino acids”. Therefore, I suggest the authors to check the chromatograms and re-edit the sequences, but those gaps shouldn’t have to be there.

Experimental design

The research question as stated in the introduction and methods are clear. And the authors manage to answer the questions properly. However, the very fact that the sequence alignment has problems, as pointed out above, start to raise question about the quality of the study.
I suggest the authors to revise the chromatograms and detect what is the problem with those gaps and fixed, this will probably require looking and editing the original sequences.

Validity of the findings

In this section I return to same issue related to the alignment.

Additional comments

Suggestions in the text.
Line 46 add Mexico, after Quintana Roo
Line 56 it is estrange that a group of smaller fishes represent a large number of the biomass
Line 77-78 revise English
Line 82-83 consider revising the construction of this sentence
Line 97 consider change the word “works” for studies or papers
Line 100-101 what field is poorly explored?
Line 102 replace “knowing” for to know
Line 112 change “in the” for at
Line 127-128 weighting samples or measuring samples multiple times is a common practice, normaaly, you would takes several measurements and at the end used the mean value of the measurements taken, was that what you did? If this is what it was done, it needs to be clarified in the methods.
Line 133, 164,
Line 182-187 revise this paragraph at the end is hard to follow.
Line 193 Base pair normally is abbreviated bp, please switch.
Figure #4 in the circle for Gear for Adults, change to Cast nets and Dip nets
Figure #6 switch the axis; biomass should be the dependent variable and abundance the independent variable.

Annotated reviews are not available for download in order to protect the identity of reviewers who chose to remain anonymous.

Reviewer 3 ·

Basic reporting

Uh-Navarrete et al. describe the fish community composition of the Chile Verde Lagoon in the Yucatan Peninsula of Mexico from samples of adult, juvenile and larval fishes collected in five sampling trips during 2022. Fishes were collected with a combination of fishing gears that included nets and traps. Fish identification was done using traditional taxonomy for the adults and juvenile specimens and using DNA barcodes for the larval fishes. The authors found a total of 27 fish species with a dominance in abundance of the Clupeidae and Gobiidae families. They highlight the presence of a marine gobid in their samples (Microgobius microlepis), a species that they claim has never been observed in a freshwater system. The manuscript concludes stating that the contribution provides a baseline for future environmental assessments of the lagoon.

The contribution of Uh-Navarrete et al. seems to be the extension of work done to document the freshwater fish diversity in a unique group of lagoons in the Yucatan Peninsula (e.g. Uh-Navarrete et al. 2021 Diversity). The authors highlight the use of DNA barcodes as a useful method to correctly identify larval stages which would be otherwise difficult to identify. This is nicely exemplified in Uh-Navarrete et al. 2021 and in this contribution.

I feel that this manuscript needs a careful revision of the English language to make several points clearer throughout the manuscript (I have provided a few examples where clarity is lacking, see other comments).

The paper does not have a research question as objective (or a hypothesis) and is rather trying to document the fish diversity in the Chile Verde Lagoon. If this is supposed to be a more ecological type of paper, I think that the manuscript needs much more context about the setting of the lagoon and the environment around it. The literature referenced seems to be adequate for what is known about freshwater fishes in that area. I wonder if references to any other similar system outside Mexico (if any), would provide a better context of this research.

The manuscript is structured in a conventional way. I have suggested minor changes to the structure and the figures in the additional comment box.

Experimental design

The methods section would also need to provide more detail about the study site, sampling design, etc, to be able to be replicated. The DNA barcoding methods used seem to be standard to what is normally used in this kind of studies.

Validity of the findings

I have checked the supplements provided by authors and those include the DNA barcode data as well as more detail about sampling and experimental design. I wonder if some of these supplements could be included in the manuscript to make clear the sampling intensity and the data used to make the species accumulation curves presented there. In the Conclusions, it would be good to provide a wider context of the findings regarding what has been found in adjacent lagoons of that area. Again, if a similar system outside the Yucatan Peninsula exists, the discussion and conclusion could be enriched by referring to those studies.

Additional comments

Specific comments

Title

- Change the “richness of fishes” for “fish species richness”
- Mention that the study was conducted in Mexico

Abstract.

L46. Include Mexico at the end of Quintana Roo
L48. Change the comma after gears for a period and start a new sentence there.

Introduction

L67-69. This argument does not explain why there are more freshwater fishes in the tropics

L80-82. I do not think that the generalization that “works that consider eggs and larvae found more taxa” is valid. The completeness of fish inventories depends more on the sampling method and sampling intensity.

L84-86. Be more specific when saying that there are no taxonomic keys for freshwater fishes. Perhaps you are referring to you study area or region.

L97-99. Avoid giving the exact number of studies here. By the time you ms is published, that number will be already outdated.

L109. It should be “larval fishes” and not “larvae fish”. Also check throughout the manuscript the use of fish vs fishes. “The plural of more than one individual of a single species is ‘fish’, but it is ‘fishes’ if there is more than one species”

Materials and Methods

I would expect to have a better description of this system which seems to be unique. Is it connected to other nearby lagoons or to the sea? I have seen papers from the co-authors of this manuscript where the description of the system is provided (e.g. for Lake Bacalar). It would be good to have such description here.

L112-113. How many people were involved in sampling? How many traps, seines and cast nets were used? This information is needed.

L118. An “m” is missing after the µ.

L127. Explain here the method to calculate the biomass. Referring to that reference is not sufficient. Also, can you specify how many plankton samples (with the two methods) were collected for sampling trip and per sampling point?

Results

L185-186. Italicize genera and species names

L208-216. Revise the use of commas and periods here. Some sentences have to be separated by periods.

L223. If the name of this species is provided for the first time they have to be spelled out. Check that also in the next lines.

L250-259. Move this section to the beginning of the Results

Discussion

L367. Be aware that the definition of a fish nursery area is complex and I do not think that you have sufficient information to suggest that this lagoon is a nursery area. See: Beck, et al., 2001. The identification, conservation, and management of estuarine and marine nurseries for fish and invertebrates:. Bioscience, 51(8), pp.633-641.

Figure 3. It is not possible to understand which samples were used to build this species accumulation curve. The caption needs to specify this.

Figure 6. I do not understand this graph. To which abundance and biomass are they referring to? Fig 6B why not showing the mean and SD from each sampling date. In that way, it would be easier to make comparisons.

---

## Round 0.2 · Minor Revisions

I dont know why you want to insist Chile lagoon is freshwater
It is not: a psu of 2.2 is not freshwater, regardless of what is producing the salinity.
To gain a better understanding I discussed this water quality issue with a fish biologist who is very familiar with your area. He said Chile Lagoon, Bacalar and lagoons like them are called "salinas", because they are not freshwater (note that one of the references you cited, Alcocer & Escobar, called such things "inland saltwaters"). Based our my conversation with that person it is clear that Chile lagoon is not saline due to marine input as its well connected to Bacalar (which is saline due to non-marine input) AND the flow of water in the channel between Chile Lagoon and Chetumal Bay is outwards from the lagoon to the sea, not the reverse. That person also said salinity in Chile lagoon is a bit higher than in Bacalar because Chile is shallow and more subject to evaporation.
So you need to change wording relating to that question; and you could include something about what the direction of flow is indicating about the source of the salinity. This paper is simply not acceptable until that change is made, to reflect the reality of the salinity situation in Chile Lagoon. Editing to address that will not be difficult to do.

The reviewer also made useful points that must be addressed too.
There are good, free websites, such as https://www.grammarly.com/, that can help with English construction, which does need attention.

You have the basis of a good paper, so take these items into account when revising the manuscript.

**Language Note:** The Academic Editor has identified that the English language must be improved. PeerJ can provide language editing services - please contact us at [email protected] for pricing (be sure to provide your manuscript number and title). Alternatively, you should make your own arrangements to improve the language quality and provide details in your response letter. – PeerJ Staff

Reviewer 1 ·

Basic reporting

The English grammar has not improved, it is presently not acceptable. I attach a list of example line-by-line wording changes I suggest, but I could not annotate the entire manuscript - but it gives some ideas of how to improve the writing- best to have it gone over by a fluent English speaker. The words are correct but the arrangments of almost every sentence is not standard English.

Literature references are still excessive - bringing 4 down to 3 or 2 is not enough - there are often two refs for a statement when one is enough..esp, when it is a distant example (like a Brazil plus a Yucatan).. often a second one is just added for no reason, other than filler, like line 131- the Ko paper is fine and addresses the difficulty of ID, why add Sheraliev, an obscure barcoding study from Uzbekistan?

Experimental design

The authors do not explain how a species was identified from a BIN code- what if there are two names in the BIN? what if there are other BINs labeled with that name, why do you believe the ID from an unknown submitter? It is all about vouchers AND your evaluation of the correctness of the name for the BIN. Did you have your own vouchers? of course.

Please read and cite https://zookeys.pensoft.net/article/83795/

for explaining what to do with a BIN name..

concerning the paragraph on line 406- no they are not all published-- species identification is not standardized, any name is accepted by BOLD-- BINs cannot "verify" an ID-- not based on "data generated over time", an ID is based on evaluation of the vouchers and sources and photographs and biogeography. Note that if you have your own vouchers, that is a critical piece of validation.

Validity of the findings

The Opsanus FASTA is still listed as Opsanus in the raw data- it is Batrachoides gilberti

Additional comments

The authors need to more closely follow the recommendations from the prior round of review- there is a sign this rewrite was rushed- in fact, after telling the authors that if more than one species of fish is discussed, one needs to say "fishes". In fact, in some places the revised version replaces correct fishes and inserts fish instead- which is puzzling.



note the line numbers are from the revised ms version- please make sure to use the ms that fits the line numbers here.

47- changed fishes to fish? whenever more than one species is referred to, as a noun, it is fishes. Better to say fish fauna, or ichthyofauna.

49- using mtDNA barcodes (first time use barcodes need to clarify it is DNA)

50 "a variety of sampling gear"

51 supported is not standard English for identifications- "obtained using"

52- in general, abundance is singular..

53- "the latter" means excess sentences--when adding a minor point, no need for extra sentences- just parentheses "biomass (wet weight, suction technique) were calculated from 43 samples".

57- corresponds, occupied and represented are all not standard English as used-- use "comprised" for percentages of a total and "accounted for" for "what proportion"

59- no values.. just greatest biomass

58- total abundance, not abundances- make sure to search and replace "abundances"

60- sentences are too long and with extraneous words ( "tropical"?).. 'Morphology was sufficient to identify adult and juvenile fishes; however, larval stages usually could not be identified to species. For eggs and larvae, DNA barcoding is necessary to identify species."

67- not "initial".. a basis is a basis

72- FISHES, please do a search and replace for fish to fishes;

71- "the" for tropical regions, I cannot explain why English does this.

72- why are you replacing fishes with fish?-- it is always fishes if more than one species (unless an adjective)- do search and replace

117- "especially in terms of biodiversity"-- no "The".. just Freshwater fishes

121- remove long wordy phrases not needed- like "to mention some of the common problems that have the most", just say "have a significant impact..."

123 discovered instead of known- Costello Tedesco and Lee refs all cover the same subject-- one is enough. This is just background philosophy.

125- no "of them" , it is already said to be species.

126-- well, we can protect unknown species-- the main reason to conserve whole ecosystems. I would leave out that phrase.

127- "Inventories of ichthyofauna usually focus on adult fishes, however some, for example Schmitter.. have shown that including eggs and larvae can significantly increase species numbers"

130- long sentences can be shortened, no need to add fish when it is understood etc. "Early stages are often not included in inventories..".. ID is "difficult".. complex is not an obstacle (i.e. all statistics are complex).

132- delete Sheraliev.. and reduce the whole next sentence down to a phrase added on to the prior sentence", especially for freshwater..."

133 scarce

138 independent

139 not fish, you mean species

140 fishes!!!!

141-- three refs for this simple statement- Brazil, Brazil, and Yucatan.. just use Yucatan. There are 100 refs you could use, so just use a local one.

142- "Some authors consider that molecular identification is necessary to corroborate" -- those two refs are a random pick for this statement.. just say "DNA identification is often necessary to corroborate..."

144- recognize not used for stages, use identify. I would not enumerate the number of studies as "no more than 30"- you have not searched the entire world's literature. Just say "relatively few" and only 4 for mexico (that one you can be sure)

146- knowledge is not a field.. and US FW fishes are well documented-- use "The early life history of neotropical freshwater fishes remains poorly documented, but is of increasing interest due to the threats to these ecosystems and the vulnerability of eggs and larvae to environmental changes."

150 - present tense-- we document.. the richness of the fish fauna.. and establish a baseline species inventory including early stages identified using DNA barcodes

248 - a massive

250- 2002). The age.. no "the" before Creatceous

253- It is 26

257- to the southeast

258- no the for Chetumal Bay

261- cannot say no studies for the system when Bacalar is part of the systems- no studies specifically including the Chile Verde system, however Bacalar Lake is reported to have elevated...

265- not according to.. that is for a person.. "Based on the chemistry of these freshwater..."

267 vs. sodium

273 on field trips

275 FISHES

291- never say as follows when you can just say it.. "Fish larvae were collected..."

296 fixed in.. transported on ice... stored at -18.. (a week? just say how stored forever)

305 "Fish larvae were separated in the 15.." "and biomass was calculated as wet weight from suctioned liquid (Zavala..)" delete interstitial water,, it is in Zavala etc.

308 weighed

313- a fish that is obvious is not "keyed out"-- just say "following Schmitter..."

314- In the case of larvae is words for no reason-- just say "fish larvae were counted and sorted"

315 sorted to morphotypes based on shape...

316 instead of rays, say meristic counts

317 "in the absence of taxonomic keys for freshwater fishes in this region, we used Richards (2006) and Beeching etc..."

320 "Species names followed eschmeyer.."

383 selected for DNA sequencing

386 no to do this "as follows" just say what you did. "For larvae > 3mm...

389 "rinsed with ethanol"

390 delete first half

399 not results, "extracts were sequenced by Eurofins.."

404 remove in addition

406- no they are not all published-- species identification is not standardized, anything is accepted-- BINs cannot simply verify an ID, the IDs in BOLD need to be evaluated-- not on "data generated over time", it is based on evaluation of the vouchers and sources and photographs and biogeography. For common well sampled fishes, like snappers for example, yes, the BOLD majrity ID is correct, but that breaks down quickly with more obscure and less sampled fishes- where multiple IDs, or majority IDs are incorrect. The trouble with the toadfish is a clear example- very few samples, incorrect IDs in BOLD, and some species not barcoded at all.

---

## Round 0.3 · accepted · Accept

This is a lot better now. I have read through it myself and think it is ready to go.